# Broad Bactericidal Activity of the *Myoviridae* Bacteriophage Lysins LysAm24, LysECD7, and LysSi3 against Gram-Negative ESKAPE Pathogens

**DOI:** 10.3390/v11030284

**Published:** 2019-03-21

**Authors:** Nataliia P. Antonova, Daria V. Vasina, Anastasiya M. Lendel, Evgeny V. Usachev, Valentine V. Makarov, Alexander L. Gintsburg, Artem P. Tkachuk, Vladimir A. Gushchin

**Affiliations:** 1N.F. Gamaleya Federal Research Centre for Epidemiology and Microbiology, Ministry of Health of the Russian Federation, 123098 Moscow, Russia; northernnatalia@gmail.com (N.P.A.); d.v.vasina@gmail.com (D.V.V.); evgenyvusachev@gmail.com (E.V.U.); gintsburg@gamaleya.org (A.L.G.); artem.p.tkachuk@gmail.com (A.P.T.); 2Lomonosov Moscow State University, 119991 Moscow, Russia; kazejosei@gmail.com; 3Center for Strategic Planning of the Ministry of Health of the Russian Federation, 119435 Moscow, Russia; makarovvalentine@gmail.com

**Keywords:** bacteriophages, *Myoviridae*, bacteriophage-derived lytic enzyme, enzybiotics, endolysin, in vitro activity, ESKAPE

## Abstract

The extremely rapid spread of multiple-antibiotic resistance among Gram-negative pathogens threatens to move humankind into the so-called “post-antibiotic era” in which the most efficient and safe antibiotics will not work. Bacteriophage lysins represent promising alternatives to antibiotics, as they are capable of digesting bacterial cell wall peptidoglycans to promote their osmotic lysis. However, relatively little is known regarding the spectrum of lysin bactericidal activity against Gram-negative bacteria. In this study, we present the results of in vitro activity assays of three putative and newly cloned *Myoviridae* bacteriophage endolysins (LysAm24, LysECD7, and LysSi3). The chosen proteins represent lysins with diverse domain organization (single-domain vs. two-domain) and different predicted mechanisms of action (lysozyme vs. peptidase). The enzymes were purified, and their properties were characterized. The enzymes were tested against a panel of Gram-negative clinical bacterial isolates comprising all Gram-negative representatives of the ESKAPE group. Despite exhibiting different structural organizations, all of the assayed lysins were shown to be capable of lysing *Pseudomonas aeruginosa*, *Acinetobacter baumannii*, *Klebsiella pneumoniae*, *Escherichia coli*, and *Salmonella typhi* strains. Less than 50 μg/mL was enough to eradicate growing cells over more than five orders of magnitude. Thus, LysAm24, LysECD7, and LysSi3 represent promising therapeutic agents for drug development.

## 1. Introduction

Antibiotic microbial resistance (AMR) is a natural aspect of microbe evolution under selective pressure. Because most antibiotics currently in use have natural analogues with similar native structures, AMR-associated genes of environmental bacteria can easily be distributed to clinically important strains through horizontal gene transfer [1,2]. The misuse, such as for the treatment of viral infections; underuse (premature antibiotic treatment termination); and overuse of antibiotics in agriculture leads to the catastrophic spread of AMR among bacteria surrounding human habitats [3,4,5,6,7]. According to a World Health Organization (WHO) report [3], the greatest attention needs to be paid to pathogens of the ESKAPE group (*Enterococcus faecium*, *Staphylococcus aureus*, *Acinetobacter baumannii*, *Pseudomonas aeruginosa*, *Klebsiella pneumoniae*, and other Enterobacteriaceae species). These pathogens represent the greatest threat among the so-called superbugs, which can rapidly acquire resistance to several classes of antibiotics and are able to cause a variety of nosocomial infections, such as bacteremia, pneumonia, and wound and skin infections [3,8].

Bacteriophages were used to control microbial populations long before penicillin was discovered [9]. The revived interest in bacteriophages and their enzymes as antibacterial agents is an expected consequence of entry into the so-called “post-antibiotic era,” when most efficient and safe antibiotics will not provide a sufficient level of defense against bacterial pathogens. The use of bacteriophages has many advantages [10]. However, virus-based drugs also have significant drawbacks, including their labor-intensive acquisition and associated stringent safety requirements; also, they are unstable during storage and need constant pharmacokinetic monitoring [11,12]. Moreover, bacteriophages are highly immunogenic agents, which can reduce the effectiveness of their repeated use [9,10,11]. The primary problem in the application of bacteriophage-based therapeutics is their unpredictable action toward patient-specific infections, necessitating the performance of additional in vitro tests prior to treatment.

Drugs based on bacteriophage enzymes, such as lysins, rather than the viruses themselves, can have much more predictable results. Bacteriophage lysins are enzymes encoded by bacteriophages that cleave peptidoglycans (PGs) in the cell walls (CW) of bacteria. During their life cycle, bacteriophages secrete lysins to deliver phage DNA into cells during bacterial infection as well as to release new virions from the cell [13]. The catalytic domains of bacteriophage lysins can possess different classes of PG-degrading activities, including transglycosylase, glucosaminidase, lysozyme-like, amidase, and endopeptidase activities [14]. Among the benefits of lysins are their high rate of lytic action, their ability to act upon antibiotic-resistant bacterial strains (including bacteria growing under different metabolic conditions), as well as their degradation of bacterial biofilms [15] and the low probability of the development of bacterial resistance to lysins. A wide spectrum of endolysin activities, comparable to modern antibiotics, would solve one of the primary problems associated with bacteriophage use. It could allow therapies to be compatible with the condition of patients with acute infections, and the treatment outcome would be more predictable under conditions when there is a lack of time for preliminary in vitro testing.

In contrast, endolysins are traditionally believed to have lytic activity against the specific hosts of the phages from which they were isolated. Thus, individual endolysins are proposed to act against specific strains and species or, rarely, more broadly against the genera of Gram-negative bacteria [16,17]. Thus, most studies have focused on identifying the effects of these enzymes and their modified variants against the host species of parental phages [16,18,19]. However, the conserved PG structure of Gram-negative bacteria make the application endolysins potentially useful with respect to their breadth of action [20,21,22]. Recently, additional evidence has demonstrated the activity of specific endolysins toward Gram-negative bacteria [23,24], at least for prophages of *Escherichia coli* and *A. baumannii*. However, few studies have described the broad activity of bacteriophage-encoded lysins against Gram-negative bacteria, and growing evidence suggests that many bacteriophage lysins are capable of much more than just acting against parental phage-specific hosts.

To study the activity of lysins against a wide spectrum of Gram-negative bacteria, we cloned three putative endolysins from *Myoviridae* bacteriophage family members (LysAm24, LysECD7, and LysSi3) and studied their activity extensively in vitro. Bacteriophages encoding the selected lysins belong to different genera of the *Myoviridae* family. Importantly, the selected enzymes represent lysins with diverse domain organization (single-domain vs. two-domain) and different predicted mechanisms of action (lysozyme-like vs. peptidase). All of the enzymes were affinity purified and tested under different in vitro conditions against a panel of Gram-negative clinical bacterial isolates, including Gram-negative representatives of the ESKAPE group of pathogens (*P. aeruginosa*, *A. baumannii*, *K. pneumoniae*, *E. coli*, and *Salmonella typhi* strains). All three assayed lysins exhibited a wide spectrum of activity against the assayed bacteria, and less than 50 μg/mL of each enzyme was sufficient to eradicate growing cells of the tested bacterial strains over more than five orders of magnitude.

## 2. Materials and Methods

### 2.1. Bacterial Strains

The bacterial strains used in the study included both reference strains and clinical isolates of Gram-negative representatives of the ESKAPE group of pathogens, including *A. baumannii*, *P. aeruginosa*, *K. pneumoniae*, *E. coli*, and *S. typhi* as well as *Staphylococcus haemolyticus* and *S. aureus* from the collection of the N.F. Gamaleya Federal Research Center for Epidemiology and Microbiology, Ministry of Health of the Russian Federation (Table 1). Three strains of *K. pneumoniae* (B3060, Osh-2k, I6208) were from the state collection of pathogenic microorganisms and cell cultures “GKPM-Obolensk.” All of the strains were stored at –80 °C and cultivated in LB broth at 37 °C and 240 rpm overnight before the assays were performed.

### 2.2. Recombinant Expression and Purification of Proteins

The coding sequences for the selected endolysins were PCR amplified from inactivated phage lysates (kindly provided by Dr. Eugenie O. Rubalskii) and were subsequently cloned into the expression vector pET42b(+) (kanamycin resistance) and checked for errors via Sanger sequencing (for primers, see Appendix A). All of the proteins contained a C-terminal 8-His tag for affinity purification. The expression vectors were introduced into the competent *E. coli* cells, strain BL21(DE3) pLysS (chloramphenicol resistance) using a heat shock transformation protocol.

The *E. coli* cells were grown in LB broth (37 °C, 240 rpm) to an OD_600_ value of 0.55–0.65 and then induced with β-d-1-thiogalactopyranoside (1 mM IPTG) at 37 °C for 3 h. The cells were harvested by centrifugation (6000× *g* for 10 min at 4 °C) and resuspended in lysis buffer (20 mM Tris HCl, 250 mM NaCl, and 0.1 mM EDTA, pH 8.0), incubated with 100 µg/mL lysozyme at room temperature for 30 min, and disrupted by sonication. The cell debris was removed by centrifugation (10,000× *g* for 30 min at 4 °C) and the supernatant was filtered through a 0.2 µm filter. The proteins were purified on an NGC Discovery^тм^ 10 FPLC system (Bio-Rad, Hercules, CA, USA) with a 5 mL HisTrap FF column (GE Healthcare, Chicago, IL, USA) pre-charged with Ni^2+^ ions. The filtered lysate was mixed with 30 mM imidazole and 1 mM MgCl_2_ and loaded on the column that was pre-equilibrated with binding buffer (20 mM Tris HCl, 250 mM NaCl, and 30 mM imidazole, pH 8.0). The fractions were eluted using a linear gradient to 100% elution buffer (20 mM Tris HCl, 250 mM NaCl, and 500 mM imidazole pH 8.0). The collected protein fractions were dialyzed against 20 mM Tris HCl (pH 7.5). The purity of the proteins was determined by 16% SDS-PAGE.

The protein concentrations were measured using a spectrophotometer (Implen NanoPhotometer, IMPLEN, Munich, Germany) at 280 nm and calculated using a molar extinction coefficient of E^0.1%^_280nm_.

### 2.3. Antibacterial Assay

Overnight bacterial cultures (OD_600_ = 1.4–1.6) were used as stationary phase cultures or were diluted 30-fold in LB broth and grown to exponential phase (OD_600_ = 0.6). Subsequently, the cells were harvested by centrifugation (3000× *g*, 10 min) and resuspended in the same volume of 20 mM Tris HCl (pH 7.5). Each suspension was diluted 100-fold in the same buffer to a final density of approximately 10^6^ cells/mL. Afterwards, 100 µL of the bacterial suspension and 100 µL of the protein at the appropriate concentration were mixed in 96-well plate wells, and buffer without endolysins was used as a negative control. The mixtures were incubated at 37 °C for 30 min with shaking at 200 rpm and then were diluted 10-fold in PBS (pH 7.4). Subsequently, 100 μL of each dilution was plated onto LB agar, and bacterial colonies were counted after overnight incubation at 37 °C. All of the experiments were performed in triplicate, and the antibacterial activity is expressed as log10 of the number of surviving bacterial colonies.

The spectrum of antimicrobial activity was tested against a panel of sixteen Gram-negative bacterial strains and two Gram-positive bacterial strains (Table 1) using the conditions described above.

The effects of pH, salts, and buffers (Na or K phosphate buffers, pH 7.5) on the specific activity of endolysins were analyzed using the *A. baumannii* strain Ts 50-16 cultured to the logarithmic growth phase. The bacteria were incubated with the proteins in 20 mM Tris HCl buffer with different pH values (5.0 to 9.0); 5, 10, or 50 mM of Na or K phosphate buffer (pH 7.5) and 5 mM of Na or K phosphate buffer (pH 7.5) supplemented with different NaCl or KCl salts (0 to 500 mM). The effect of EDTA on the bactericidal activity of lysins at different pH values was assessed as mentioned above after the addition of 0.5 mM of permeabilizer during cell incubation with the lysins.

### 2.4. Storage Stability

To investigate the storage stability of the endolysins, the proteins were exposed to different temperature conditions (4, –20, and –80 °C) for one week or one, two, or three months. The protein samples were stored in 20 mM Tris HCl, 100 mM KCl, and 50% glycerin at pH 7.5. Antimicrobial activity was assessed as described above using the *A. baumannii* strain Ts 50-16.

### 2.5. Dynamic Light Scattering

The hydrodynamic diameters of the protein particles in solution were measured using a ZetaSizer Nano-ZS (Malvern Instruments LTD, Malvern, UK) in polystyrene cuvettes with an optical path length of 1 cm and a laser wavelength of 633 nm. The protein samples (approximately 1 mg/mL) were assayed in 10 mM sodium phosphate buffer (pH 7.5). The samples were heated from 25 to 70 °C with 2.5 °C step increases. The data were analyzed using Dispersion Technology version 5.10 (Malvern, PA, USA).

## 3. Results

### 3.1. Sequence Analysis, Expression, and Physicochemical Properties of Recombinant Enzymes

In this study, three putative endolysin-encoding genes from lytic phages of the family *Myoviridae* were assayed. The genes encoding LysAm24, LysECD7, and LysSi3 from Acinetobacter phage AM24 (NCBI: txid1913571), which infects *A. baumannii*, *Escherichia* phage ECD7 (NCBI: txid1981499, *E. coli*) and Enterobacteria phage UAB_Phi87 (NCBI: txid1197935, *S. typhi*), respectively, were cloned and expressed in *E. coli* (Table 2). None of these endolysins had been previously cloned or characterized.

BLAST searches for the deduced amino acid sequences derived from the cloned nucleotides showed a strong similarity to the phage-related lysozyme-like superfamily (muramidases, GH24 family) for the LysAm24 and LysSi3 proteins, whereas LysECD7 contained a d-alanyl-d-alanine carboxypeptidase domain (peptidase M15 family). Furthermore, whereas LysECD7 and LysSi3 contained only one enzymatic catalytic domain (ECD), the deduced amino acid sequence of LysAm24 included an additional CW-binding domain (CBD) at the N-terminus and an ECD at the C-terminus (Figure 1a). Such an inverted orientation of the CBD and ECD has been shown to be characteristic of Gram-negative bacteriophage endolysins [25,26].

To clone the selected lysins, their coding sequences were amplified from inactivated phage lysates. Subsequently, the amplified fragments were fused to an 8-His tag at the C-terminus of the encoded protein and were inserted into the *E. coli* expression vector pET42b(+). The proteins were purified using NiNTA affinity chromatography followed by SDS-PAGE gel analysis, which showed the monomeric form of the individual proteins with apparent molecular masses of 25.9, 16.1, and 18.5 kDa for LysAm24, LysECD7, and LysSi3, respectively (Figure 1b).

The use of dynamic light scattering allowed the hydrodynamic diameter as well as the thermal and structural stability of proteins molecules to be evaluated. The results showed that hydrodynamic diameters for all three enzymes had narrow peaks corresponding to 5.29 ± 0.97, 3.94 ± 1.33, and 3.69 ± 0.87 nm for LysAm24, LysECD7, and LysSi3, respectively (Figure 1c), indicating that all of the recombinant molecules are stable as primarily monomers in solution.

Figure 1d presents the results showing the dependence of the light scattering intensity parameters on temperature. The observed sharp increase in the total light scattering intensity and in the hydrodynamic diameter of the enzymes particles indicates the beginning of the aggregation of the proteins. The aggregation of LysAm24, LysECD7, and LysSi3 was observed to begin at 55, 50, and 42.5 °C, respectively. Thus, the results showed that LysAm24 possesses slightly increased thermostability compared to LysECD7 and LysSi3.

### 3.2. Bactericidal Activity and Biochemical Properties of Recombinant Lysins

The bactericidal activity of endolysins varies significantly depending on the protein of interest and the bacterial species and strains used. According previous studies, the optimal concentrations for endolysins vary from 10 to 500 µg/mL [24,27,28]. To initially evaluate bactericidal activity, the *A. baumannii* strain Ts 50-16 (for LysAm24 and LysSi3) and the *E. coli* strain M15 (for LysECD7) were used. The bactericidal activities of LysAm24, LysECD7, and LysSi3 against exponentially growing bacterial cells were shown to be concentration-dependent (Figure 2a). The minimal active concentration after 30 min of incubation in 20 mM Tris HCl buffer (pH 7.5) was observed to be 0.5 µg/mL for LysAm24 and LysECD7 and 10 µg/mL for LysSi3. Specifically, reductions of over 2 and 1.35 logs, respectively, were observed after treating the bacterial cells with 0.5 µg/mL of LysAm24 and LysECD7, and a reduction of approximately 3 logs was observed after the cells were treated with 10 µg/mL of LysSi3. In general, LysSi3 exhibited 10-fold lower activity than LysECD7 and LysAm24 against the selected bacterial strains. Lysin concentrations of more than 5 μg/mL (LysAm24 and LysECD7) and 50 μg/mL (LysSi3) eliminated bacterial growth completely, suggesting that recombinant LysAm24, LysECD7, and LysSi3 can exert bactericidal action without additional membrane permeabilization.

It is a known fact that lysins are generally less active against stationary-phase bacteria compared to exponentially growing bacterial cells. Indeed, for *A. baumanii* Ts 50-16, the addition of 10 μg/mL LysAm24, LysECD7, or LysSi3 reduced the viability of exponentially growing cells (OD_600_ = 0.6) by 76%, 70%, and 80% respectively, while reducing the viability of stationary-phase bacteria (OD_600_ = 1.4–1.6) by 47%, 33%, and 23% only. Even a concentration of endolysins of 100 μg/mL did not allow complete elimination of bacteria (Figure 2b). However, the addition of 0.5 mM EDTA restored the activity of lysins, indicating that the problem is related to the ability of the enzymes to overcome the bacterial outer membrane but not bactericidal activity itself.

For the effective use of endolysins, it is essential that they retain their activity under physiological conditions (high salt content, pH close to the physiological values present in the blood, oral cavity or on skin (7.4, 7.0, and 5.5, respectively), and exhibit resistance to proteolytic cleavage). Thus, the effect of the composition of the buffer system used and the salts present in the solution (Figure 2) on the preservation of the activity of endolysins was assessed.

We evaluated the effects of the presence of salts (NaCl or KCl) on the activity of the endolysins (Figure 2c). Both KCl and NaCl had an inhibitory effect on lysin activity, and the presence of both salts reduced the activity of all three assayed enzymes. The activity of LysAm24 was decreased by almost two-fold in the presence of 25 mM NaCl, while KCl had the same effect at 50 mM. For LysECD7, 100 mM of either salt significantly decreased bactericidal activity, while the activity of LysSi3 was inhibited in the presence of 50 mM NaCl or KCl. In general, all three lysins tolerated the presence of KCl better than NaCl.

Different buffer systems, specifically sodium or potassium phosphate buffers, at various concentrations and at pH 7.5 were also investigated, as these solutions are likely to be more suitable and cost-effective for future animal and human applications than buffers containing Tris HCl. The results of assays evaluating the effects of phosphate buffers at concentrations of 5, 10, and 50 mM on the activity of the enzymes showed that the most appropriate concentration was 5 mM, with potassium–phosphate buffer (pH 7.5) being optimal for LysAm24 and sodium–phosphate buffer (pH 7.5) being optimal for LysECD7 and LysSi3.

In addition, the stability of the enzymes during stock storage in the presence of 100 mM KCl and 50% glycerol at three temperatures (+4, −20, and −80 °C) for three months was tested. Bactericidal activity was tested in Tris HCl buffer pH 7.5. All of the preparations retained their initial activity after up to two months of storage. However, after three months, the activity of LysAm24 was almost undetectable, while that of LysECD7 and LysSi3 decreased significantly (for all three enzymes, the inhibition of the growth of bacterial colonies was only 1- and 0.5-orders of magnitude of that of the control, respectively, rather than 5-orders of magnitude). These results indicate that storage condition optimization is necessary for these enzymes.

### 3.3. Effect of EDTA on the Bactericidal Activity of Lysins at Different pH Values

A separate series of experiments was carried out to investigate the activity of LysAm24, LysECD7, and LysSi3 over a broad range of pH values in vitro (Figure 3), the results of which showed that all three enzymes had similar bactericidal activity in Tris HCl buffer at different pH values. The highest activity of LysAm24, LysECD7, and LysSi3 was observed under moderate acidic conditions (pH = 5.0–6.0). The activity of all three enzymes decreased when the pH was increased to 6.5–7.0, with LysAm24, LysSi3, and LysAm24 exhibiting 45%, 30%, and 60% of their initial activity, respectively. At higher pH values (7.5–8.0), the activities of all three enzymes recovered, albeit to different degrees.

The effect of EDTA, which is used as a membrane permeabilizer to enhance the bacteriolytic activity of lysins, was also evaluated (Figure 3). The synergy between EDTA and the enzymes was observed under neutral and alkaline conditions (pH 6.5–8.5). The most pronounced effect of the presence of EDTA was observed for LysAm24 and LysSi3, for which almost the complete recovery of activity was observed at pH values of 6.5–7.0, whereas this effect occurred to lesser degree for LysECD7.

### 3.4. Broad Substrate Specificity and Antibacterial Properties of Recombinant Phage Endolysins

The spectrum of activity of the three endolysins was tested against a panel of eighteen bacterial isolates from patients, including *A. baumannii*, *P. aeruginosa,* and *K. pneumoniae* as well as a collection of reference strains (*E. coli* and *S. typhi*) (Table 1). The results of the antimicrobial assays of the three endolysins towards these bacterial species are presented in Figure 4. All of the enzymes showed a wide but diverse range of bactericidal activity and efficiently inhibited the growth of several strains each from *P. aeruginosa*, *A. baumannii*, and *K. pneumoniae* (Figure 4). When the proteins were added to exponentially growing cultures of bacteria, no growth of the host cells was observed within 18 h of the treatment. However, the strains *P. aeruginosa* Ts 44-16 and *K. pneumoniae* F 104-14 were only moderately susceptible to the lysins. The strain *E. coli* BL21(DE3) pLysS, when used for the recombinant expression of lysins, was partly resistant to action of LysAm24 and LysSi3. Additionally, LysSi3 had no activity against the *K. pneumoniae* B3060, Osh-2k, and I6208 strains. Among the Gram-positive bacterial strains, no activity was found against *S. aureus* 73-14, but LysAm24 and LysSi3 partially inhibited the growth of *S. haemolyticus* G58 (to 45% and 10%, respectively).

Interestingly, although all three assayed lysins were active or moderately active against the *E. coli* BL21(DE3) pLysS expression strain, extensive cells lysis of these cells was not detected during the expression and purification procedures, indicating that either internal cell membrane penetration did not take place or that lysis of the expression strain occurred to some degree with no significant effect on the efficiency of protein production.

## 4. Discussion

All three phage endolysins assayed in this study were obtained from members of the family *Myoviridae* that differ in their taxonomic assignments: Escherichia phage ECD7, Tevenvirinae subfamily; Enterobacteriaphage UAB_Phi87, Ounavirinae; and Acinetobacter phage AM24, different unclassified genus. The significance of the multiple-drug resistance of these phage hosts is reflected in the recent classification of these species as “Priority 1: Critical” pathogens on the WHO priority pathogens list for the R&D of new antibiotics [3]. The genomes of the selected bacteriophages are highly variable in terms of both size and organization. Since most bacteriophages of this family are lytic, they have potential use as antibacterial agents by themselves. Taking into account the size of the cloned lysins in this study (25.9, 16.1, and 18.5 kDa for LysAm24, LysECD7 and LysSi3, respectively), we expected them to be endolysins, because virion-associated lysins (VALs) have molecular weights of greater than 40 kDa [13]. Small endolysins are compatible with cost-effective biotechnological production, and our results show that LysAm24, LysECD7, and LysSi3 can be efficiently and cheaply produced in *E. coli*. Furthermore, it was shown that the assayed enzymes have good stability profiles and possess wide bactericidal activity, as they were capable of inhibiting the growth of all assayed Gram-negative representatives of the ESKAPE group. The bactericidal effect against Gram-positive bacterial strains was not pronounced. No activity was found against *S. aureus* 73-14, while LysAm24 and LysSi3 partially inhibited the growth of *S. haemolyticus* G58 (to 45% and 10% respectively). More strains of Gram-positive bacteria need to be tested to assess the activity of the studied lysins on this group more carefully.

The endolysins chosen for this study represent proteins with diverse domain organizations, including those with the most common catalytic domains; glycosidases, which hydrolyze glycosidic bonds in glycan strands; and endopeptidases, which cleave the cross-bridges between strands and are proposed to have distinct mechanisms of bactericidal action. This set of endolysins allowed us to study the general patterns of activity in this class of proteins.

Endolysins from phages infecting Gram-positive bacteria are believed to not exhibit broad lytic activity and are typically limited to particular bacterial genera, species, or even strains [29,30,31]. This is probably due to the highly diverse structures of the CW peptidoglycans of Gram-positive bacteria needed to evade the host immune system [20,21,22]. Thus, the bacteriophage lysins from a Gram-positive background may require tailoring to the CW, leading to a narrowing of their spectrum of specificity. In addition, bacteriophages infecting Gram-positive bacteria need to control the activity of lysins released from newly lysed cells to prevent the premature lysis of neighboring cells, which are suitable for infection by the released bacteriophage progeny. The firm fixation of endolysins on the CW debris of lysed Gram-positive bacterial cells could prevent their activity toward the CW surface of neighboring cells. This explanation seems to be feasible [13] and is supported by CBD domains being most commonly present in the endolysins of phages that infect Gram-positive bacteria and mycobacteria [14,32]. It was previously shown that the omission of the CBD in Gram-positive endolysins significantly improves their activity and broadens the range of their bacterial targets [33,34]. Thus, the presence of specific CBD and diverse PG structures makes the activity of Gram-positive endolysins narrowly targeted [35,36].

In contrast, Gram-negative bacteria possess an outer membrane (OM) which explains the conservative nature of Gram-negative bacterial PG structures [20,21,22]. Endolysins of Gram-negative bacteria are small molecules that commonly have a single-domain globular structure. With some exclusions, lysins that act against Gram-negative bacteria do not have a CBD domain [25,37]. Thus, because the absence of a CBD domain in targeting endolysins is the norm for Gram-negative bacteria, such lysins should generally be much less specific.

Some experimental evidence of the broad lytic activity of endolysins against Gram-negative bacteria has come from studies of *E. coli* and *A. baumannii* prophages [23,24,27]. An *E. coli* prophage lysin (PlyE146) has been shown to lyse a wide range of Gram-negative bacteria (*K. pneumoniae*, *A. baumannii*, *P. aeruginosa*, *E. coli*, and *S. enterica*) [24], and *A. baumannii* prophage lysin (AcLys) exhibits a similar range of bactericidal activity [23]. Our results show that the wide bactericidal activity of endolysins from Gram-negative-infecting bacteriophages is a much more general phenomenon than was previously appreciated. In the present study, we demonstrated that three different lysins (LysAm24, LysECD7, and LysSi3) from *Myoviridae* bacteriophage family members that exhibit diverse domain organization (single-domain vs. two-domain) and different putative mechanisms of action (lysozyme vs. peptidase) have equally wide activity profiles. Notably, even LysAm24, which contains an additional glycan-binding domain that is typically associated with strain-specific lytic activity, showed a broad spectrum of activity with respect to the assayed bacteria. A quantity of 50 μg/mL of lysins was enough to eradicate growing bacterial cells by over more than five orders of magnitude for the assayed *P. aeruginosa*, *A. baumannii*, *K. pneumoniae*, *E. coli*, and *S. typhi* strains. In contrast, PlyE146 [24] only displayed wide specificity at a concentration of 400 μg/mL. Moreover, similar results were achieved at lower concentrations of this enzyme (below 1 μg/mL) only after cells were partially permeabilized by hypotonic conditions, freezing during storage or after the cells were washed with water [23].

Importantly, none of the recombinant enzymes assayed in our study (LysAm24, LysECD7, and LysSi3) required a specialized OM penetration approach to exert full bactericidal activity at an acidic pH and low salt concentration on exponentially growing cells. We previously showed that the recombinant endolysin L-KPP10 only exhibits pronounced activity in the presence of 0.5 mM EDTA [28] and that L-KPP10 alone does not possess significant lytic activity even on exponentially growing cells. Thus, EDTA and organic acids that displace divalent cations and destabilize the bacterial cell membranes could be an optional solution for the use of such endolysins [13]. Alternatively, the addition of a positively charged peptide (SMAP-29) at the C-terminus of the endolysin L-KPP10 (generating AL-KPP10) eliminated the need for EDTA permeabilization [28]. The effectiveness of this approach has been demonstrated in several other studies [16,17], suggesting that the use of positively charged peptide fusion may be a more convenient way to ensure efficient outer membrane permeabilization.

Interestingly all three assayed lysins were active or moderately active against the *E. coli* BL21(DE3) pLysS expression strain used for the production of recombinant enzymes (Figure 4). The strain was fully sensitive to LysECD7 (100%) and partly resistant to the action of LysAm24 (70%) and LysSi3 (10%). Moreover, all proteins were shown to be soluble when expressed intracellularly. In our previous studies of L-KPP10 endolysin fused with the strong cell penetration peptide SMAP-29 [28], we were able to purify it from inclusion bodies (insoluble condensed matter). All attempts to produce it in a soluble form led to either a lack of protein expression or the production of a soluble protein with no bactericidal activity. Lysins described in the current paper do not possess specific membrane penetration peptides. It is not clear whether studied endolysins can be obtained from the cytoplasm, where recombinant proteins are expressed, into periplasmic space where they can exert their bactericidal action. On the one hand, there is evidence that, for efficient permeabilization, some lysins require additional phage-encoded proteins for the control of inner membrane permeabilization [14]. For the lysins under study, the necessity of additional membrane permeabilizing proteins is not known and requires further investigation. On the other hand, as we mentioned above, buffer conditions, salts, pH, and membrane permeabilizers significantly affect the bactericidal activity of lysins. Thus, the intracellular conditions might be not optimal for enzymatic action, allowing soluble endolysins to be produced intracellularly in the strain that is sensitive to its action from outside.

The finding that none of our enzymes required a chemical permeabilizer under some conditions was unexpected. Some lysins are known to possess a natural ability to penetrate the bacterial OM due to positively charged segments that can disorganize or disrupt the OM [13,23], and the recombinant enzymes assayed in this study likely possess such sequences. Specifically, the C-terminal sequences of LysAm24 and LysSi3 contain a positively charged α-helix with nine lysine and arginine residues in 30-aa stretch. For proteins with homologous domain structures, this region was speculated to be an OM penetration sequence [24]. In contrast, no such positively charged stretch of amino acids is present in LysECD7. In ChiX, an L-Ala-D-Glu endopeptidase from *Serratia marcescens* that is homologous to LysECD7, the formation of a sugar-binding site presumably occurs through three tryptophan residues (Trp87, Trp89, and Trp116) that occupy the outer edge of the active site cleft [38]. However, LysECD7 only contains two of these residues (Trp80 and Trp105), and whether these amino acids possess permeabilization activity is not yet clear. An alternative explanation for the cell permeabilization activity of the recombinant enzymes assayed in this study is that they contain an 8-His tag sequence at the C-terminus with adjacent sequences that form a positively charged cluster that is capable of improving membrane penetration (KLEHHHHHHHH). To test this hypothesis, we performed assays over a range of pH values with and without added permeabilizer (EDTA).

The observed pH-dependent changes in the dynamics of the assayed endolysin activities indirectly confirmed this hypothesis. Decreased activity in response to increasing pH values is known to affect other endolysins [24]. The additional protonation of C-terminal cationic peptides of LysAm24 and LysSi3 increase their destabilizing activity toward the bacterial OM at low pH values. Indeed, the charges of LysAm24 and LysSi3 C-termini changed from +12.8/13.8 at pH 5.0 to 1.8/2.5 at pH 9.0. Since LysECD7 does not contain similar peptides, with the exception of the His-Tag sequence at the C-terminus, and the change of its C-terminus charge from +8.0 at pH 5.0 to −1.0 at pH 9, this effect was less pronounced for LysECD7, and it retained up to 65% of its activity. Under pH values with C-termini charges of less than 10.0, LysAm24 and LysSi3 activity decreased significantly in the absence of EDTA. Detailed information on the influence of different pH values on the protein charge and activity is summarized in Table 3.

To differentiate the permeabilization ability from the bactericidal activity of the endolysins, the effect of pH in the presence of EDTA, which was used as a membrane permeabilizer, was evaluated. In the presence of EDTA, the activity of all lysins increased under neutral and basic pH conditions (Figure 3). These results indicate that the decrease in activity of the proteins associated with changes in pH is primarily associated with the ability of the lysins to overcome the cell membrane rather than by changes in their bactericidal activity. Our results suggest that there is a correlation between the structural changes in proteins under different pH values and their endogenous ability to penetrate the bacterial membrane (Spearman’s rank correlation coefficients *r* = 0.86, 0.73, and 0.53 for LysAm24, LysECD7, and LysSi3 correspondingly). The charges of the C-termini can explain the observed changes but not the recovery of their activity over a narrow range of physiological pH values (approximately pH 7.5). Further detailed structural studies of these proteins will shed light on this issue.

The bactericidal activity of all studied lysins was concentration-dependent with one exception. At the highest concentration point (100 μg/mL), LysAm24 had two times weaker activity on the stationary phase bacterial cells (Figure 2) compared to at the lower concentration (10 μg/mL). For LysSi3 with putative muramidase activity, as well as for LysECD7 with putative peptidase activity, this effect was not observed. This is an interesting fact that we suggest is characteristic of some endolysins. Previously, we have shown the same effect for endolysin with putative N-acetylmuramidase activity—artificial lysin AL-KPP10 fused with antimicrobial peptide SMAP-29 and tested against *P. aeruginosa* [28]. It was significantly less active at a concentration of 100 μg/mL than at concentrations of 25 and 50 μg/mL. The possible explanation for this lysin activity drop off under high concentrations is that some lysins might be prone to self-aggregation (e.g., dimerization of molecules). Interestingly, the presence of EDTA fully compensated for this drop in activity.

The presence of potassium and sodium chlorides negatively affected the activity of all studied endolysins. Interestingly, the bactericidal activity of LysAm24 and LysSi3 was restored to up to 60–80% that of the control at a concentration of 250 mM of either salt. For LysECD7, this effect was less pronounced and seemed to move to a higher salt concentration, at least for potassium chloride. Proteins usually have a particular optimal salt concentration because salt affects the intermolecular interactions. Thus, this effect can hardly be explained by the properties of the recombinant proteins alone. The bactericidal activity was tested on a complex system that included live bacterial cells. To exclude the factors pertaining to the live bacterial cells, experiments with pure peptidoglycan would be very useful. The specificity of bacteriophages and their encoded endolysins is considered to be a benefit for their practical use and preparation [16,17]. However, we believe that endolysins with activity toward specific bacteria are disadvantageous, as this would make patient treatment outcomes less predictable and would likely require preliminary sensitivity evaluations. Our results show that a large number of bacteriophage lytic enzymes can have broad bactericidal activity toward Gram-negative bacteria, demonstrating their potential in the development of medicines with reliable and predictable effectiveness.

## Figures and Tables

**Figure 1 viruses-11-00284-f001:**
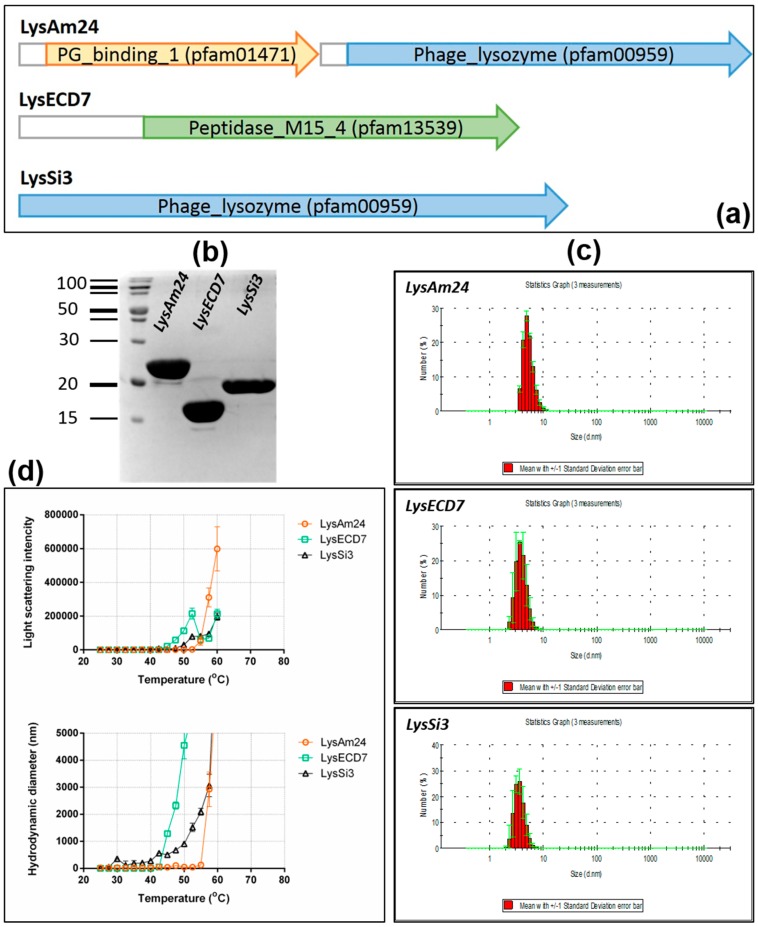
Domain organization, purification, and physicochemical properties of LysAm24, LysECD7, and LysSi3. (**a**) Putative domain organization of LysAm24, LysECD7, and LysSi3 predicted from the deduced amino acid sequences. The prediction was done with the protein BLAST search (https://blast.ncbi.nlm.nih.gov/Blast.cgi). (**b**) SDS-PAGE gel analysis of purified endolysins. The PageRuler Broad range Unstained Protein Ladder (Thermo Scientific, Vilnius, Lithuania) was used (**c**) Evaluation of the hydrodynamic diameter of *E. coli*-produced endolysins by DLS analysis. Statistical distribution of particle size by number. The data are presented as the mean values of three measurements ± SD. (**d**) Temperature dependence of the total light scattering intensity (upper panel) and the hydrodynamic diameters of the particles (lower panel). The data are presented as the mean values of three measurements made with an interval of 15 s ± SD.

**Figure 2 viruses-11-00284-f002:**
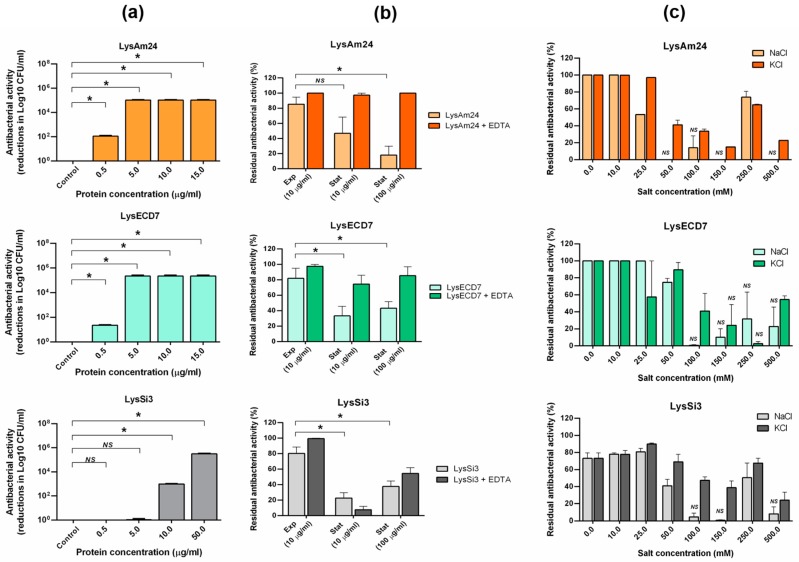
Bactericidal activity and biochemical properties of LysAm24, LysECD7, and LysSi3. (**a**) Bactericidal activity of different concentrations of LysAm24 against *A. baumannii* Ts 50-16, LysECD7 against *E. coli* M15 and LysSi3 against the *A. baumannii* Ts 50-16 in exponential phase. Cell cultures without incubation with endolysins were used as controls. The number of surviving cells after 18 h is expressed as reduction in log10 CFU/mL compared to the control. NS, no statistical significance of the data compared to the untreated culture is observed (*p* > 0.05, Mann–Whitney test). (**b**) Activity of lysins against both exponentially growing cells (Exp) and stationary-phase cells (Stat) of *A. baumannii* Ts 50-16 without and in the presence of EDTA. The residual activity after 18 h of growth compared to the untreated culture is shown. NS, no statistical significance of the data compared to the exponential growth phase is observed (*p* > 0.05, Mann–Whitney test). (**c**) Effects of salts on the bactericidal activity against *A. baumannii* Ts 50-16. The residual activity after 18 h of growth compared to the untreated culture is shown. NS, no statistical significance of the data compared to the untreated culture is observed (*p* > 0.05, Mann–Whitney test). For all experiments, the mean values are shown (± standard error of the mean (SEM)) from three independent experiments. Asterisk (*) indicates significant effect on bactericidal activity.

**Figure 3 viruses-11-00284-f003:**
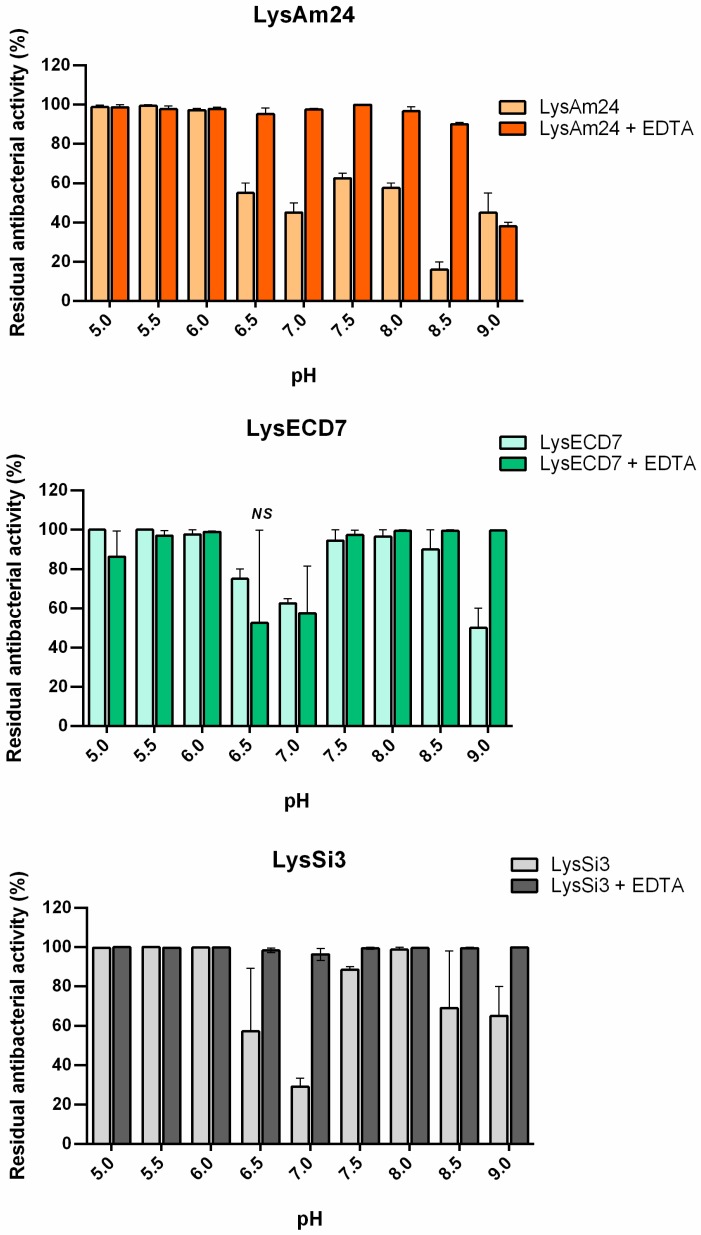
Effects of pH and EDTA on the antibacterial activity of LysAm24, LysECD7, and LysSi3 against exponentially growing cells. The residual activity after 18 h of growth compared to the untreated culture is shown. For all experiments, the mean values are shown (±SEM) from three independent experiments. NS, no statistical significance of the data compared to the untreated control culture (*p* > 0.05, Mann–Whitney test). All other data points were statistically significant compared to the untreated control culture.

**Figure 4 viruses-11-00284-f004:**
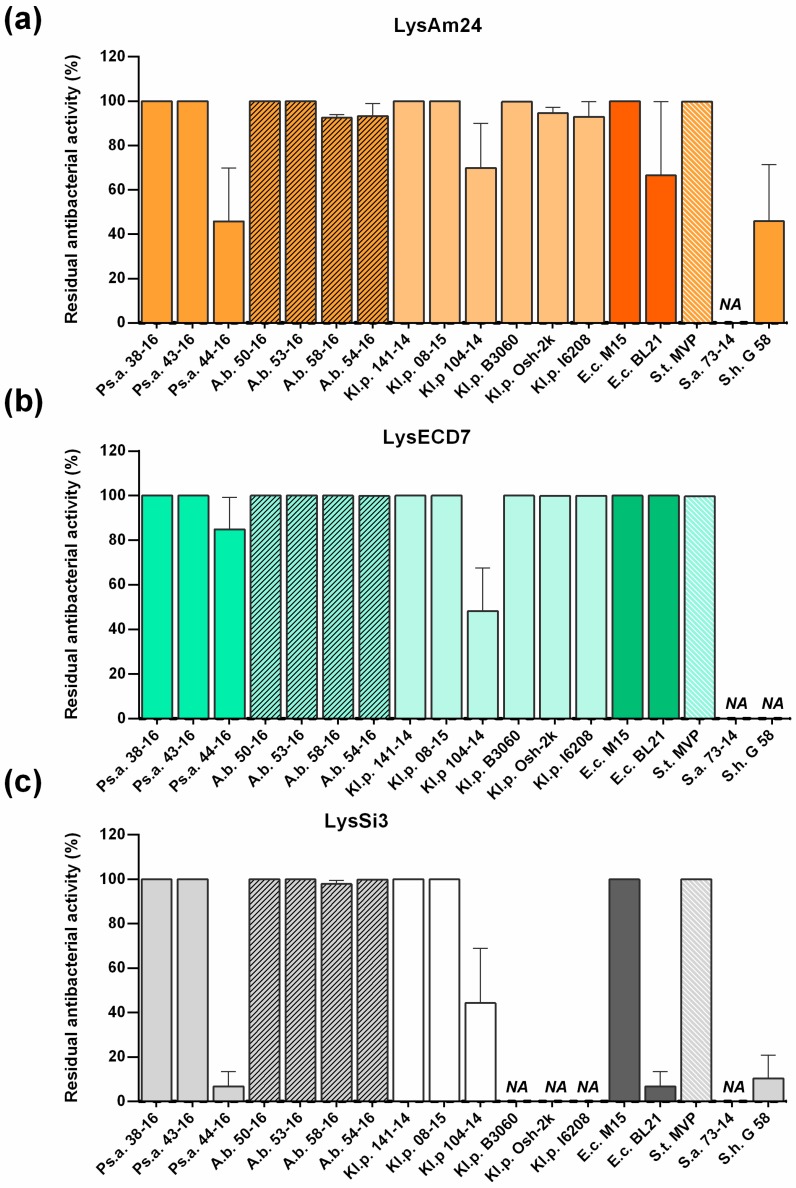
Endolysin activity against different strains of Gram-negative bacteria. Bacterial count (cfu/mL) was used as the method of choice for all strains: (**a**) LysAm24, (**b**) LysECD7, (**c**) LysSi3. For all experiments, the mean values are shown (±SEM) from three independent experiments. NA, no bactericidal activity was detected. All other data points were statistically significant compared to the untreated control culture (*p* < 0.05, Mann–Whitney test).

**Table 1 viruses-11-00284-t001:** Bacterial strains used in the study and the sources of their isolation.

Strain	Source	
*Pseudomonas aeruginosa* Ts 38-16	Patient’s sputum, hospital strain	ICU ^1^
*Pseudomonas aeruginosa* Ts 43-16	Patient’s wound fluid, hospital strain	ICU
*Pseudomonas aeruginosa* Ts 44-16	Patient’s urea, hospital strain	ICU
*Acinetobacter baumannii* Ts 50-16	Patient’s sputum, hospital strain	ICU
*Acinetobacter baumannii* Ts 53-16	Patient’s sputum, hospital strain	ICU
*Acinetobacter baumannii* Ts 58-16	Infusion pump, hospital strain	ICU
*Acinetobacter baumannii* Ts 54-16	Infusion pump, hospital strain	ICU
*Klebsiella pneumoniae* Ts 141-14	Patient’s urea, hospital strain	Inpatient hospital
*Klebsiella pneumoniae* Ts 08-15	Patient’s sputum, hospital strain	ICU
*Klebsiella pneumoniae* F 104-14	Patient’s sputum, hospital strain	Outpatient hospital
*Klebsiella pneumoniae* B3060	Patient’s liquor, hospital strain	Inpatient hospital
*Klebsiella pneumoniae* Osh-2k	Patient’s blood, hospital strain	ICU
*Klebsiella pneumoniae* I6208	Patient’s oral pharynx, hospital strain	Inpatient hospital
*Escherichia coli* M15	Reference laboratory strain	-
*Salmonella typhi* MVP 728	Reference laboratory strain	-
*Staphylococcus aureus* Z 73-14	Patient’s nasal swab	Outpatient hospital
*Staphylococcus haemolyticus* G 58-0916	Patient’s urethra	Outpatient hospital
*Escherichia coli* BL21(DE3) pLysS	Laboratory expression strain	-

^1^ ICU: intensive care unit.

**Table 2 viruses-11-00284-t002:** Endolysins used in the study.

Enzyme	Enzyme Source	Phage Host	GB Accession No.
LysAm24	Acinetobacter phage AM24	*A. baumannii*	APD20282.1
LysECD7	Escherichia phage ECD7	*E. coli*	ASJ80195.1
LysSi3	Enterobacteria phage UAB_Phi87	*S. typhi*	YP_009150069.1

**Table 3 viruses-11-00284-t003:** Heatmap of estimated endolysin C-termini charges over the pH range and the residual antibacterial activity within this range (http://protcalc.sourceforge.net/). Min to Max pKa and residual activity is indicated with the color scale from red to green background. For all experiments, the mean values of three independent experiments are shown.

pH	C-Termini Charge (pKa)				Residual Antibacterial Activity, %		
	LysAm24	LysECD7	LysSi3		LysAm24	LysECD7	LysSi3
5.00	12.80	8.00	13.80		98.8	100.0	99.8
5.50	11.60	7.30	12.60		99.,4	100.0	100.0
6.00	10.20	6.10	11.20		97.2	97.6	99.9
6.50	8.00	4.00	9.00		55.0	75.0	57.2
7.00	5.80	1.80	6.80		45.0	62.5	29.2
7.50	4.40	0.50	5.40		62.5	94.5	88.5
8.00	3.40	−0.30	4.40		57.5	96.5	98.8
8.50	2.60	−0.70	3.50		16.0	90.0	69.0
9.00	1.80	−1.00	2.50		45.0	50.0	65.0

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
