# Peer review of "Broad Bactericidal Activity of the Myoviridae Bacteriophage Lysins LysAm24, LysECD7, and LysSi3 against Gram-Negative ESKAPE Pathogens"

_viruses, 2019, doi:10.3390/v11030284_

Round 1
Reviewer 1 Report
In this study, three putative phage lysins of Gram-negative bacteria was expressed and were tested against several clinical isolates of bacteria. However, this study is in a preliminary stage. This manuscript contained too little research content. Much more experiments should be carried out, such as key sites of lysins, the mechanism of these lysins, the protective effect of lysins in vivo.
minor:
1. The authors should have someone with a good command of English review the manuscript.
2. The Introduction section is too long.
3. The number of bacterial strains used in the study is too few. Each bacterial species of Gram-negative representatives of ESKAPE group only contain two isolations. Much more isolations of each bacterial species should be used to determine the lytic spectrum of these lysins. Gram-positive bacterial species should also be used to determine the lytice spectrum of lysins.
4. Line 119: why 100 μg/ml lysozyme was incubated with E. coli cells?
5. Did the imidazole in the elution buffer has been removed in the purified lysins? The imidazole has antibacterial activity against Gram-negative bacterium.
6. The calcium/zinc dependent of lysins should also be detected.
7. The Result section contained too much discussions.
8. Lines 224-225: “LysAm24, LysECD7 and LysSi3 can effectively penetrate the outer membrane without additional permeabilization and reach the substrate”. LysAm24, LysECD7 and LysSi3 show lytic activity, indeed suggestting that they can penetrate the outer membrane of bacteria. This manuscript indicated that these lysins are soluble expression. Why they can’t penetrate the cell membrane of bacteria and lyse the E. coli BL21 when these lysins were expressed? Since these lysins show lytic activity against Escherichia coli М15. Did these purified lysins also show bactericidal activity against E. coli BL21???
9. Line 438: References 21.
10. Figure S1 is too vague.
Author Response
Point 0: In this study, three putative phage lysins of Gram-negative bacteria was expressed and were tested against several clinical isolates of bacteria. However, this study is in a preliminary stage. This manuscript contained too little research content. Much more experiments should be carried out, such as key sites of lysins, the mechanism of these lysins, the protective effect of lysins in vivo.
Response 0: The key point of the paper is to provide first systemic data about generally broad bactericidal activity of lysing from Gram-negative bacteriophages. Three lysins were expressed and extensively studied in vitro. Physic and chemical properties, role of buffer conditions, salt, pH, permeabilizer, lysin concentration and storage conditions has been studied. Additionally, 18 strains representing all members of ESCAPE group had been used to show broad lytic activity. Thus, the subject of this article is the study of the physic and chemical properties of three new, not previously described endolysins. Key sites of lysins, the mechanism of action, the protective effect in vivo will be studies in frame of future research.
Point 1: The authors should have someone with a good command of English review the manuscript.
Response 1: We have handled the English language and style editing of the manuscript. It was edited by the Nature English Editing Service.
Point 2: The Introduction section is too long.
Response 2: Both reviewers 2 and 3 indicated that the introduction provide sufficient background and include all relevant references. Thus, we decided to keep most of the content.
Point 3: The number of bacterial strains used in the study is too few. Each bacterial species of Gram-negative representatives of ESKAPE group only contain two isolations. Much more isolations of each bacterial species should be used to determine the lytic spectrum of these lysins. Gram-positive bacterial species should also be used to determine the lytice spectrum of lysins.
Response 3: We conducted additional experiments on more Gram-negative and Gram-positive bacterial species and made the suggested corrections to the text.
Point 4: Line 119: why 100 μg/ml lysozyme was incubated with E. coli cells?
Response 4: This is a standard method of bacterial cell lysis before purification of expressed proteins. Affinity purification eliminates lysozyme residues.
Point 5: Did the imidazole in the elution buffer has been removed in the purified lysins? The imidazole has antibacterial activity against Gram-negative bacterium.
Response 5: Thank you for the pointing it out. The imidazole in the elution buffer has been removed. The collected protein fractions were dialyzed against 20 mM Tris HCl pH 7.5 buffer. Spectra of purified enzymes did now have signs of imidazole.
Point 6: The calcium/zinc dependent of lysins should also be detected.
Response 6: We have been doing additional experiments in presence of calcium/zinc but we will not be able to provide this data for current manuscript.
Point 7: The Result section contained too much discussions.
Response 7: Thank you. Most of discussion moved to discussion section.
Point 8: Lines 224-225: “LysAm24, LysECD7 and LysSi3 can effectively penetrate the outer membrane without additional permeabilization and reach the substrate”. LysAm24, LysECD7 and LysSi3 show lytic activity, indeed suggestting that they can penetrate the outer membrane of bacteria. This manuscript indicated that these lysins are soluble expression. Why they can’t penetrate the cell membrane of bacteria and lyse the E. coli BL21 when these lysins were expressed? Since these lysins show lytic activity against Escherichia coli М15. Did these purified lysins also show bactericidal activity against E. coli BL21???
Response 8: Thank you. We have added particular piece to results “Interestingly, although all three assayed lysins were active or moderately active against the E. coli BL21(DE3) pLysS expression strain, cells lysis of these cells was not detected during the expression and purification procedures, indicating that either bacterial cell membrane penetration did not take place or that lysis of the expression strain occurred to some degree. Although the assayed proteins are putative endolysins, these results indicated their potential failure to pass through the bacterial inner membrane from within the cell, a possibility that requires additional studies.”
Point 9: Line 438: References 21.
Response 9: Thank you. Corrected now.
Point 10: Figure S1 is too vague.
Response 10: Thank you. We have decided to remove this picture. It does not improve paper.
Reviewer 2 Report
The antimicrobial activity of recombinant putative phage lytic enzymes with C-terminal-fused His-tag, were investigated. They showed antimicrobial activity against Pseudomonas aeruginosa, Acinetobacter baumannii, Klebsiella pneumoniae, Escherichia coli and Salmonella typhi strains. Less than a 50 μg/ml were enough to decrease the number of living cells for five.
Major comment
Authors measured an antibacterial activity treated by recombinant phage lytic enzymes. In the manuscript, authors also used terms “lytic activity” and “enzyme activity”, but they are not the correct expression. The experiment to observe cell lysis and peptidoglycan hydrolysis should be needed.
In addition, authors mentioned the effect of C-terminal cationic peptide for the cell wall lytic activity without treatment of permeabilizing-reagents in discussion, even the effect of C-terminal fused his-tag for the antibacterial activity has not been evidenced. To emphasize the relatively higher antibacterial activity of these recombinant phage lysins to Gram-negative bacteria, this point is critical. If C-terminal fusion of his-tag to recombinant phage lysin increases the antimicrobial activity against Gram-negative bacteria, it would provide the new application to drug development.
Minor comments
1) Legend of figure 1: Sentences for b) and c) should be exchanged.
2) Lanes 252-258: Authors described that 5 mM buffers are “optimal”, but they should use “highest” because they did not test at lower buffer concentration.
3) Lanes 269-272: Are these stabilities almost same under the temperatures tested?
4) Lane 321: Authors said, "In contrast, Gram-<positive> bacteria", but it should be <negative>.
5) Lane 438: Second “21” should be removed.
Author Response
Point 1: The antimicrobial activity of recombinant putative phage lytic enzymes with C-terminal-fused His-tag, were investigated. They showed antimicrobial activity against Pseudomonas aeruginosa,Acinetobacter baumannii, Klebsiella pneumoniae, Escherichia coli and Salmonella typhi strains. Less than a 50 μg/ml were enough to decrease the number of living cells for five.
Response 1: All right. The key point of the paper is to provide first systemic data about generally broad bactericidal activity of lysing from Gram-negative bacteriophages. Three lysins were expressed and extensively studied in vitro. Physic and chemical properties, role of buffer conditions, salt, pH, permeabilizer, lysin concentration and storage conditions has been studied.
Point 2: Authors measured an antibacterial activity treated by recombinant phage lytic enzymes. In the manuscript, authors also used terms “lytic activity” and “enzyme activity”, but they are not the correct expression. The experiment to observe cell lysis and peptidoglycan hydrolysis should be needed.
Response 2: In frame of the present paper we changed term “lytic activity” to “bactericidal activity”. Experiments to observe cell lysis and peptidoglycan hydrolysis are in progress and will be described in further papers.
Point 3: In addition, authors mentioned the effect of C-terminal cationic peptide for the cell wall lytic activity without treatment of permeabilizing-reagents in discussion, even the effect of C-terminal fused his-tag for the antibacterial activity has not been evidenced. To emphasize the relatively higher antibacterial activity of these recombinant phage lysins to Gram-negative bacteria, this point is critical. If C-terminal fusion of his-tag to recombinant phage lysin increases the antimicrobial activity against Gram-negative bacteria, it would provide the new application to drug development.
Response 3: Role of C-term 8his Tag in permeabilization was partially supported by experiments with EDTA addition in pH gradient. To differentiate permeabilization ability from the bactericidal activity of the endolysins, the effect of pH in presence of EDTA as a membrane permeabilizer was evaluated. In the presence of EDTA, the activity of all of the lysins increased under neutral and basic pH conditions (Figure 3). These results indicate that the decrease in activity of the proteins associated with changes in pH is primarily associated with the ability of the lysins to overcome the cell membrane rather than by changes in their bactericidal activity. Our results suggest that there is a correlation between the structural changes in proteins under different pH values with their endogenous ability to penetrate the bacterial membrane. The corresponding discussion included to the paper. More precise experiments with tag modification are in progress now.
Minor comments were corrected according to the suggestions.
Reviewer 3 Report
The current manuscript describes a finding that novel endolysins exhibit broad spectrum activity against various Gram negative ESKAPE pathogens. Although the data are interesting, the manuscript needs improvement in English (including many typos).
In addition,
1) the legend to Fig 1 is not correct. (b) and (c) should be switched.
2) Statistical significance of the data should be shown in Figures 2 and 3, by presenting p values.
3) In Fig 2, the enzymatic activity decreased at neutral pHs while it is higher at both acidic and basic pHs for all three enzymes. Considering structural change of proteins at different pHs, it is unusual. Please give some explanation in Discussion.
Author Response
Point 1: The current manuscript describes a finding that novel endolysins exhibit broad spectrum activity against various Gram negative ESKAPE pathogens. Although the data are interesting, the manuscript needs improvement in English (including many typos).
Response 1: We have handled the English language and style editing of the manuscript. It was edited by the Nature English Editing Service.
Point 2: the legend to Fig 1 is not correct. (b) and (c) should be switched.
Response 2: It has been corrected now
Point 3: Statistical significance of the data should be shown in Figures 2 and 3, by presenting p values. –
Response 3: It has been corrected now. More clarification is added to the picture’s legends.
Point 4: In Fig 2, the enzymatic activity decreased at neutral pHs while it is higher at both acidic and basic pHs for all three enzymes. Considering structural change of proteins at different pHs, it is unusual. Please give some explanation in Discussion.
Response 3: More experimental data with EDTA in pH gradient were added and explicitly discussed in the text (line 420-437).
Round 2
Reviewer 1 Report
1. In Table 3, activities of these three lysins were indicated as + high and +/- moderate. How to judge the activity as high or moderate? The bacterial count (cfu/ml) should be detected when decide the bactericidal activity of three lysins.
2. Whatever, since these lysins show high/moderate bactericidal activity against E. coli BL21 expression strain, it is inconceivable that these lysins can be expressed by BL21. After all, three lysins are soluble expression suggesting these lysin exist in the space beween cell wall and cell membrane, and the expression outputs are high. Thus, E. coli BL21 expression strain can be easilly killed by these lysins when they are expressed.
3. From Figure 2b, LysAm24 at high concentration (100 µg/ml) showed weaker bactericidal activity than them at low concentraion (10 µg/ml). It is very stange!!!
4. OD600.
5. Figure 3. Were the target bacteria used in the experiment was exponentially growing cells (Exp) or stationary-phase cells (Stat)?
6. Figure 4. What method used in this experiment? The decline of OD600 or bacterial count (cfu/ml)?
7. One of Table 3 or Figure 4 are enough to show the spectrum and bactericidal activity of lysins. Table 3 can be deleted.
Author Response
Point 1: In Table 3, activities of these three lysins were indicated as + high and +/- moderate. How to judge the activity as high or moderate? The bacterial count (cfu/ml) should be detected when decide the bactericidal activity of three lysins.
Response 1: Table 3 has been deleted according to your suggestion in Point 7.
Point 2: Whatever, since these lysins show high/moderate bactericidal activity against E. coli BL21 expression strain, it is inconceivable that these lysins can be expressed by BL21. After all, three lysins are soluble expression suggesting these lysin exist in the space beween cell wall and cell membrane, and the expression outputs are high. Thus, E. coli BL21 expression strain can be easilly killed by these lysins when they are expressed.
Response 2: Thank you for your question about the activity of lysins against E. coli expression strain BL21. Our previous model lysin kpp10 [Physical and chemical properties of recombinant kpp10 phage lysins and their antimicrobial activity against Pseudomonas aeruginosa. Antonova N.P., Balabanyan V.Yu, Tkachuk A.P., Makarov V.V., Guschin V.A. Bulletin of RSMU, № 1, с. 21-27 DOI 10.24075/brsmu.2018.010] was purified from inclusion bodies (insoluble condensed form). Most of attempts to make in soluble lead to the lack of protein expression or production of soluble lysin without visible activity. Many different cloning strategies were tested. This lysin was expressed in fusion with strong cell penetration peptide SMAP29. The lysins obtained in the in the present work do not have specialized membrane penetration peptides.
We do not know whether our proteins are capable of getting from the cytoplasm where the recombinant proteins are expressed into the periplasmic space as you suggest. As we have described in the present paper, the buffer conditions, salts, ions, pH, and membrane permeabilizers significantly affect bactericidal activity (probably by modulating the membrane penetration process). The simplest explanation is that the intracellular conditions do not allow for the inner membrane penetration. For that reason, many lysins require additional phage encoded proteins for controlled inner membrane permeabilization. For the lysins under study the necessity of additional proteins for efficient lysis from within is not known. Being very interesting, from within activity, nevertheless, lies out of the current paper scope and requires additional studies, probably together with a more detailed study of the outer membrane penetration process. We are currently conducting this research. We appreciate your interest to our data. We have described this effect in the discussion more clearly (Lines 420-439).
Point 3: From Figure 2b, LysAm24 at high concentration (100 µg/ml) showed weaker bactericidal activity than them at low concentraion (10 µg/ml). It is very stange!!!
Response 3: This is an interesting fact and as we suppose that this effect is characteristic of some endolysins. In the present study, only one of the three lysins had this property. As you can see for LysSi3 with putative muramidase activity as well as for LysECD7 with putative peptidase activity this effect was not observed. Previously we have shown the same effect for one more endolysin with putative N-acetylmuramidase activity – artificial lysin AL-KPP10 fused with antimicrobial peptide SMAP-29 and tested against Pseudomonas aeruginosa [Antonova N.P. et al. Physical and chemical properties of recombinant KPP10 phage lysins and their antimicrobial activity against Pseudomonas aeruginosa. DOI: 10.24075/brsmu.2018.010.]. See fig 3 (A). Being applied at the concentration of 100 µg/ml it was significantly worse than at the concentration of 25 and 50 µg/ml. This effect was replicated at different strains. One explanation is that observed concentration dependent changes in activity for some lysins might be the result of self-aggregation. Another observation is that presence of EDTA under the same conditions fully compensate for the concentration-dependent drop in activity. Thus, concentration somehow affect permeabilization but not the activity of lysins. Anyway, to answer the question of what happens during the interaction of a lysin and bacterial cells, it is necessary to further study the mechanism of penetration, as well as the structure of the proteins. These data will allow to study this effect in more detail in the future.
This information was added as a part of discussion (Lines 484-495).
Point 4: OD600.
Response 4: Corrected now.
Point 5: Figure 3. Were the target bacteria used in the experiment was exponentially growing cells (Exp) or stationary-phase cells (Stat)?
Response 5: We used exponentially growing cells foк the experiments. Corrected now.
Point 6: Figure 4. What method used in this experiment? The decline of OD600 or bacterial count (cfu/ml)?
Response 6: Bacterial count was used as a method of choice for all experiments, the clarification was made in the caption to the figure 4.
Point 7: One of Table 3 or Figure 4 are enough to show the spectrum and bactericidal activity of lysins. Table 3 can be deleted.
Response 7: Corrected now. Picture is in and table is out.
Reviewer 2 Report
The broad range bactericidal activity of phage lysins against ESKAPE pathogen became clear.
Mechanism for OM penetration of these lysins would becomes an interesting subject for study.
Author Response
Point 1: The broad range bactericidal activity of phage lysins against ESKAPE pathogen became clear.
Response 1: Thank you for your valuable comments.
Point 2: Mechanism for OM penetration of these lysins would becomes an interesting subject for study.
Response 2: Thank you for your suggestions. We will definitely continue our studied with more detailed experiments on OM penetration.